# Differential Metabolite Accumulation in Different Tissues of *Gleditsia sinensis* under Water Stress and Rehydration Conditions

**Jia Liu** [1,2,3,*,†], **Rui Kang** [2,3,†], **Yang Liu** [2,4], **Ke-Xin Wu** [4], **Xue Yan** [2,3], **Ying Song** [2,3], **Li-Ben Pan** [2,3] and **Zhong-Hua Tang** [2,3]

[1] Material Science and Engineering College, Northeast Forestry University, Harbin 150040, China
[2] Key Laboratory of Plant Ecology, Northeast Forestry University, Harbin 150040, China; kangr@nefu.edu.cn (R.K.); klp15ly@nefu.edu.cn (Y.L.); yanyanyan@nefu.edu.cn (X.Y.); sy123456ssy@nefu.edu.cn (Y.S.); pan7216@nefu.edu.cn (L.-B.P.); tangzh@nefu.edu.cn (Z.-H.T.)
[3] College of Chemistry, Chemical Engineer and Resource Utilization, Northeast Forestry University, Harbin 150040, China
[4] School of Forestry, Northeast Forestry University, Harbin 150040, China; wukexin94@126.com or klp16wkx@nefu.edu.cn
[*] Correspondence:klp14lj@nefu.edu.cn; Tel: +86-451-8219-2098; Fax: +86-451-8210-2082
[†] These two authors contribute equally to this work.

**Abstract:** *Gleditsia sinensis* Lam. is a woody species that can tolerate various drought conditions and has been widely used in all aspects of life, including medicine, food, cleaning products, and landscaping. However, few reports have focused on the regulatory mechanism of the drought response in *G. sinensis*. To understand the metabolic basis of the *Gleditsia sinensis* drought response**,** different tissues were subjected to a rehydration/dehydration treatment and subsequently analyzed using untargeted and targeted metabolomics profiling depending on gas chromatography-mass spectrometry (GC-MS) and liquid chromatography-tandem mass (LC-MS) analytical platforms, respectively. Eight sugars, twelve amino acids, and twenty phenolic compounds were characterized. Metabolites showing a significant increase or decrease under drought stress were considered to be the key metabolites of interest for a better understanding of the drought tolerance mechanisms**.** The GC-MS-identified compounds were shown to undergo tissue-specific regulation in response to drought stress. Moreover, the C6C3C6 and C6C3 structures were identified by LC-MS as phenolic metabolites, which revealed their drought-response association. Significant physiological parameters were measured, including overall plant development, and the results showed that antioxidant systems could not be completely restored, but photosynthetic parameters could be recovered. The results of this research provide insight into biochemical component information mechanism of drought resistance in *G. sinensis*.

**Keywords:** chemical components; physiological; abiotic stress; GC-MS and LC-MS

## 1. Introduction

Drought stress is one of the most unfavorable environmental factors limiting plant yield and biomass, and climate change is expected to exacerbate drought in the future [1]. More than 30% of the Earth's land surface is seriously affected by drought stress [2]. Drought stress in plants is a condition under which the amount of available water in the root zone is less than that required to maintain optimal growth and productivity [3]. Plants, as sessile organisms, have established specific

defense mechanisms that minimize cell damage and maintain growth and development under adverse environmental conditions [4]. Almost all biochemical components, including sugars, fatty acids, organic acids, and amino acids, are biosynthesized and play primary roles in the plant's life cycle, growth, and acceptive environmental challenge [5]. Additionally, under special environmental conditions and selection pressure, plants have developed the ability to synthesize specialized secondary compounds, such as terpenoids, alkaloids, saponins, and phenolic metabolites, that supply a competitive advantage in ecological interactions [6]. In particular, phenolic metabolites play an important role in the response to stressful environmental conditions, such as drought, temperature extremes, and high ultraviolet radiation [7]. Phenolic components are antioxidants with a wide range of functions, such as scavenging reactive oxygen species, catalyzing oxygenation reactions through the formation of metallic complexes, and inhibiting the activities of oxidizing enzymes [8]. Twenty phenolic compounds were identified in both well-watered and water deficit-stressed tobacco plants, and most of these compounds accumulated at higher levels in the control group [9]. Transcriptomic and metabolomic studies on *Arabidopsis* plants verified that positively-regulated flavonoid accumulation under drought stress greatly benefits stress resistance [10]. On the other hand, some previous reports have demonstrated the use of exogenous phenolic metabolites to increase drought resistance in plants [11]. The effects of drought stress on the diversity and density of phenolic bioactive components in the leaves of papaya have been studied to elucidate a possible defense mechanism in the papaya plant [12].

Mass spectrometry-based metabolic profiling, which examines a wide spectrum of compounds, provides a comprehensive biotechnology for investigating the metabolic level of plants under stress conditions [13]. Based on our previous research, a total of 34 types of phenolic compounds were identified in the medicinal plant *Catharanthus roseus* [14], while 149 primary metabolites were found in different tissues of ginseng species by GC-MS [15]. Recently, biochemical analyses have verified that changes in global metabolites are correlated under water deficiency in several plant species [16]. An untargeted metabolomic approach has been executed to examine differential classification compound patterns and has thus exposed some key compounds of drought response in *Arabidopsis thaliana* and *Withania somnifera* [17,18]. GC–MS metabolomics was also used to research the difference in rhizosheath weight in two switchgrass ecotypes in response to water deficiency [19].

*Gleditsia sinensis* Lam. is cultivated in most areas of China and is used as a source of natural biomacromolecules [20]. This species is a member of the legume family that is well acclimated to biotic and abiotic stresses and can tolerate drought efficiently in addition to having multiple economic uses in chemical materials, craft wood, and ornamental materials [21,22]. Recently, phytochemical research has identified functional ingredients in this species, such as triterpenoids, saponins, alkaloids, sterols, and flavonoids [23]. The extracts of these components display several important biological functions, such as antitumor, anti-inflammatory, and anti-HIV activities [24]. Over 30 compounds of triterpenoid saponins were identified in the tissues of *G. sinensis* [25]. However, these studies did not sufficiently explain the differences in bioactive metabolite strategies to improve adaptive capabilities in response to environmental factors among a variety of tissues and failed to comprehensively identify key bioactive metabolites, including phenolics, amino acids, and sugars. Therefore, in this study, we provide comprehensive bioactive compound composition and antioxidant enzyme activity profiles in response to drought stress in different tissues of *G. sinensis.* We aimed to identify chemicals that play a crucial role in the water deficit stress response of plants, which may reveal bioactive compounds to help plants combat drought.

## 2. Materials and Methods

### 2.1. Chemicals and Reagents

All chemicals and reagents (MS grade) were obtained from Fisher Scientific (Pittsburgh, PA, USA), including methanol, acetonitrile, and hexane, which were used as extraction solvents. Twenty-two phenolic metabolite standard compounds were obtained from Beijing Science and Technology (Beijing, China), including apigenin, caffeic acid, calycosin, chlorogenic acid, cinnamic acid, daidzein,

ferulic acid, galangin, genistein, isoliquiritigenin, kaempferol, *p*-coumaric acid, petunidin, *p*-hydroxycinnamic acid, liquiritigenin, *L*-phenylalanine, luteolin, naringenin, quercetin, and syringic acid. Deionized water was obtained with a Milli-Q Academic Ultra-pure Water System (Millipore, Milford, MA, USA). PEG-6000 (VWR Chemicals, Strasbourg, France) was used to induce slight to moderate water stress with the addition of 12% polyethylene glycol (PEG) (−0.2 MPa water potential) and 18% PEG (−0.4 MPa water potential) solutions, respectively.

## 2.2. Plant Materials and Growing Conditions

*G. sinensis* was grown in a greenhouse under half-day light photo cycles at a day/night temperature of 28/25 °C. Seeds were sown in pots containing perlite and maintained at 80% relative humidity until germination. Subsequently, seedlings were transferred to half-strength Hoagland solution (pH 6.5) for approximately 1 month. Seedlings were then transferred to three treatments: control, PEG 12% (medium drought stress), and PEG 18% (deep drought stress) [26]. All groups of plants were collected for metabolite analysis after 1 week of treatment. However, selected PEG 18% samples were transferred to a well-watered treatment for 1 week for the rehydration experiment. Each experiment consisted of three biological replicates.

## 2.3. Measurement of Antioxidants and Malondialdehyde

For enzyme extracts and assays, 0.2 g tissues were quenched with liquid nitrogen and then suspended in 1.6 mL of solution containing 0.1 M phosphate buffer (pH 7.4). The suspensions were centrifuged (4 °C) at 4000 rpm (10 min), and the supernatant was harvested for execution of the activities of superoxide dismutase (SOD), peroxidase (POD), ascorbate peroxidase (APX) and catalase (CAT). Malondialdehyde (MDA) was analyzed according to a previously described method [27].

## 2.4. Analysis of Photosynthesis Indexes of Plants

An automatic photosynthesis measuring apparatus (LI-6400, LI-COR Biosciences. Lincoln, NE, USA) was used to analyze the net photosynthetic rate, transpiration rate, and stomatal conductance. In addition, the measurement of chlorophyll a content in fresh leaves was conducted according to Porra et al. [28] by ultraviolet spectrophotometry (UV-2102c, Unico Instrument C, Shanghai, China).

## 2.5. GC-MS Analysis of Primary Metabolites

Primary bioactive compounds were extracted from roots, stems, and leaves by GC-MS as described by Liu et al. [15]. A 90 mg accurately-weighed sample was transferred to a 1.5 mL Eppendorf tube. Two small steel balls were added to the tube. A total of 360 µL of cold methanol (4 °C) and 40 µL of 2-chloro-l-phenylalanine (0.3 mg/mL) dissolved in methanol as an internal standard were added to each sample, and the samples were placed at −80 °C for 2 min. The mixtures were ultrasonicated at ambient temperature for 30 min, 100 Hz. Then, 200 µL of chloroform was added to the samples, the mixtures were vortexed, and 400 µL of water was added. The samples were vortexed again and then ultrasonicated at ambient temperature for 30 min. The samples were then centrifuged at 14,000 rpm and 4 °C for 15 min. Two hundred microliters of supernatant in a glass vial was dried in a freeze concentration centrifugal dryer. The quality control (QC) sample was prepared by mixing aliquots of all samples to form a pooled sample. Then, 80 µL of 15 mg/mL methoxylamine hydrochloride in pyridine was added. The resultant mixture was vortexed vigorously for 2 min and incubated at 37 °C for 90 min. Then, 80 µL of *N*,*O*-Bis(trimethylsilyl)trifluoroacetamide (BSTFA) (with 1% trimethylchlorosilane, (TMCS)) and 20 µL of hexane were added into the mixture, which was vortexed vigorously for 2 min and derivatized at 70 °C for 60 min. The samples were placed at ambient temperature for 30 min before GC-MS analysis. The GC-MS data were analyzed through an Agilent 7890A series autosampler (Agilent Technologies, Santa Clara, California, CA, USA) coupled to an Agilent 5975C gas chromatograph-mass spectrometer (Agilent Technologies, Santa Clara, California, CA, USA) and a nonpolar capillary column (30 m × 250 µm DB-5) purchased from J&W

Scientific, Folsom, CA, USA. Identical chromatogram acquisition parameters were used as outlined by Liu et al. [15]. Data were normalized by dividing each raw value by the median of all measurements of the experiment for one compound.

### 2.6. LC-MS Targeted Analysis of Phenolic Metabolites

Each tissue was divided into 0.5 g aliquots. Materials were extracted with 10 mL of 80% ethanol containing 0.1 mg/L lidocaine (internal standard), and ultrasound extraction was performed for 45 min at 45 °C and 100 Hz. Then, the extracts were filtered, and the previous step was repeated once. Finally, the filtrates were combined. Subsequently, the filtrates were centrifuged at 14,000 rpm at 4 °C (10 min), and the supernatant was collected and evaporated to dryness. The resultant extracted material was dissolved in acetonitrile (1 mL) and filtered through 0.22-μm-diameter micropores (SCAA-104, ANPEL, http://www.anpel.com.cn/). The purified solution was analyzed by LC-MS.

Analyses were executed using a ultra performance liquid chromatography (UPLC) system coupled to a time of flight mass spectrometer (QTOF) tandem mass spectrometer via an electrospray ionization (ESI) interface (Waters G2, ShangHai, Japan). Because the metabolites in plants are very complicated, it is impossible to analyze all of them with a single method. The mass spectrometer was operated in electron spray ionization (ESI$^+$) mode to analyze as many metabolites as possible in a single injection. The ESI conditions used the following parameters: capillary voltage of 3500 V, fragmentation voltage of 135 V, source temperature of 350 °C, and curtain gas pressure of 40 psi. Detection was performed in ESI$^+$ mode on a scale of m/z 50–1200. The separation was carried out on an ACQUITY UPLC BEH C18 column (1.7 μm, 2.1 mm × 50 mm; Waters, ShangHai, China) with an in-line filter at 25 °C. The injection volume was 5 μL. Gradient elution was performed at a flow rate of 0.5 mL/min with the following solvent system: (A) 0.04% acetic acid-water, (B) 0.04% acetic acid-acetonitrile; from 5% B at 0 min to 95% B at 20 min to 5% B at 22.1 min and then held at 5% B until 28 min.

### 2.7. Multivariate Analysis

The GC-MS raw data were transformed into cycling DOF factor (CDF) format with data analysis software (Agilent GC-MS 5975, Santa Clara, California, CA, USA ) and later processed by R. Each metabolite is shown as the peak area normalized to the internal standard. Peak detection, retention time alignment, and library matching were analyzed using the TargetSearch package from Bioconductor [29]. The internal standard was used for data quality control (reproducibility). Subsequently, treatment data were analyzed by SIMCA-P version 11.0 software (Umetrics, Umea, Sweden) for multivariate statistical analysis. Supervised partial least squares discriminant analysis was used to compare the treatment groups with their tissue-specific differences to identify the significant metabolites, and a *t*-test combination of approaches was used to screen the important metabolites ($p < 0.05$). A permutation test was employed to calculate the validity of the partial least-squares discrimination analysis (PLS-DA) model against overfitting, and 999 permutations were employed in all models. Hotelling's T2 region, shown as an ellipse in the score plots of the models, defines the 95% confidence interval of the modeled variation. The LC-MS data were analyzed using MassLynx version 4.1. This software detected peaks and listed the detected and matched peaks with the retention time and mass to charge ratio (*m/z*) pair and their corresponding intensities. The relative signal intensities of compounds were standardized by firstly dividing them by the intensities of the internal standard and then (max-them)/(max-mix) for which the value range is 0–1, to generate the final data matrix. The significance of the differences between means for different tissues and treatments were analyzed using Student's *t*-test and Duncan's test by SPSS 17.0. A heat map analysis was conducted with R to visualize the relative levels of phenolic compounds.

## 3. Results

### 3.1. Physiological Damage under Drought Conditions

The effects of two degrees of drought stress (12% PEG and 18% PEG) on the phenotype of plants are illustrated in Figure 1. The plants in the control group had young dark green leaves with rigid petioles, while slightly aged yellow-green leaves with fallen petioles were observed in the most severe drought treatment. As drought stress advanced, the plant growth characteristics, including the length of roots and stems and the fresh weight, were significantly lower than those of the controls (Table 1). In terms of antioxidant systems, our results show that the activity of the enzymes catalase (CAT), ascorbate peroxidase (APX), superoxide dismutase (SOD), malondialdehyde (MDA), and peroxidase (POS) in different types of tissues were significantly increased by drought, and the enzymes were specifically accumulated under severe drought conditions (Figure 1). In contrast, the response of some photosynthetic parameters (net photosynthetic rate, transpiration rate, stomatal conductance, and chlorophyll content) showed a decreasing pattern under drought stress (Figure 2).

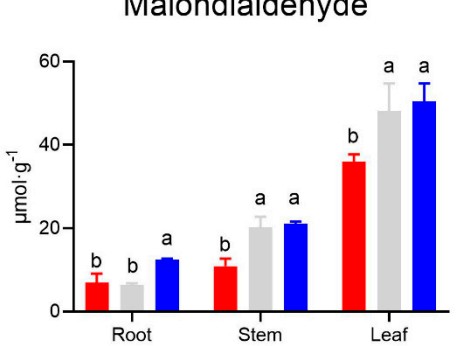

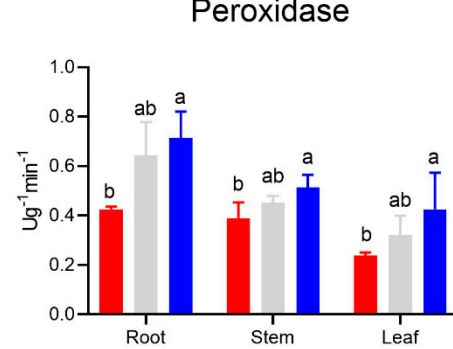

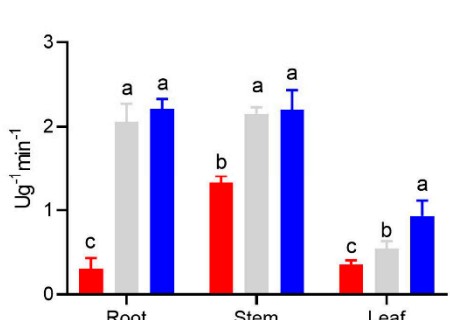

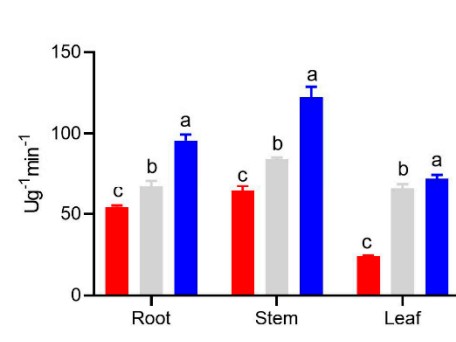

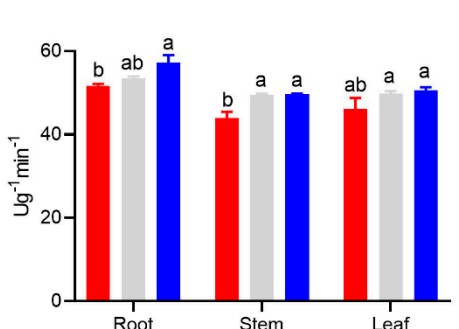

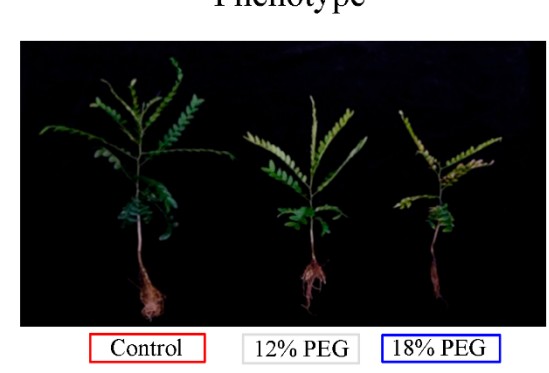

**Figure 1.** Comparison of the enzymatic activity of superoxide dismutase (SOD), peroxidase (POD), catalase (CAT), and malondialdehyde (MDA) under the application of 12% polyethylene glycol (PEG) and 18% PEG to simulate moderate and severe drought, respectively. Each bar represents the mean ± SD (*n* = 3). Different lowercase letters above the columns indicate significant differences (*p* < 0.05).

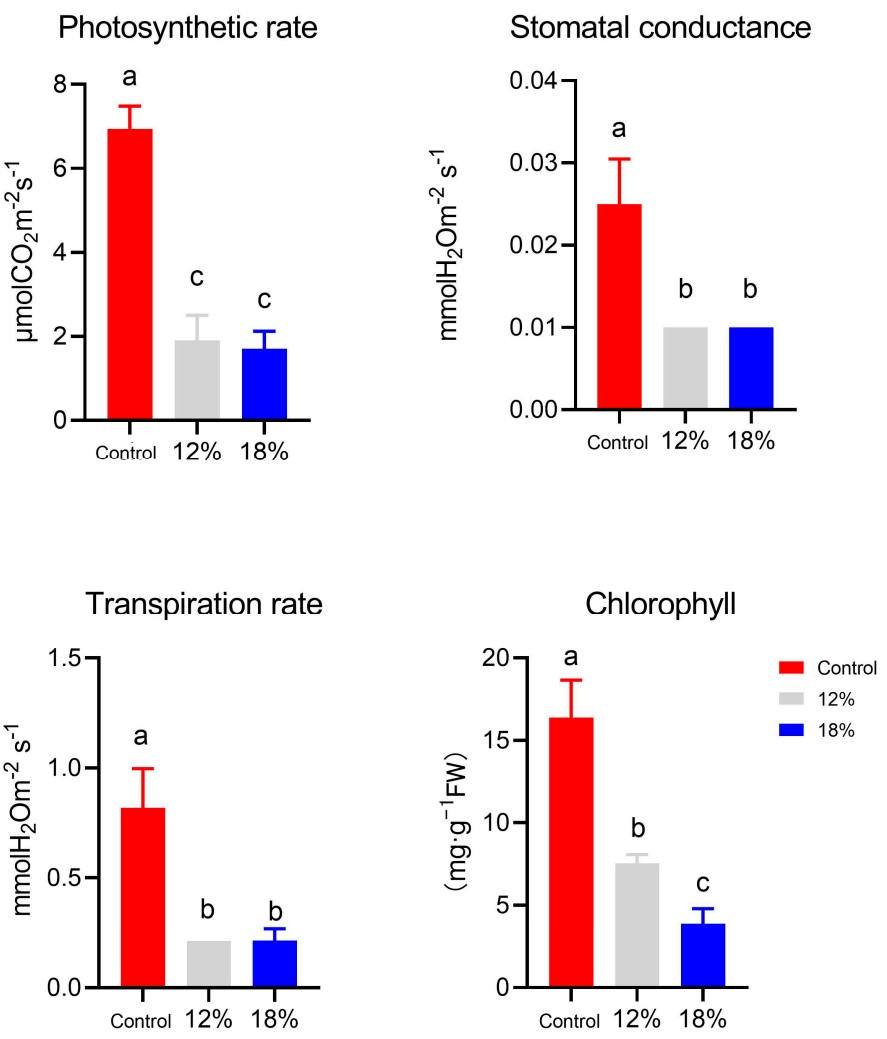

**Figure 2.** Photosynthetic parameters in root, stem, and leaf tissues of *Gleditsia sinensis* exposed to control, 12% PEG, and 18% PEG treatments. Each bar represents the mean ± SD (*n* = 3). Different lowercase letters above the columns indicate significant differences (*p* < 0.05).

**Table 1.** Plant physiological parameters of fresh weight and length. Different lowercase letters indicate significant differences (*p* < 0.05).

| Treatment | Fresh Weight (g) | | | Length (cm) | |
|---|---|---|---|---|---|
| | **Root** | **Stem** | **Leaf** | **Root** | **Stem** |
| Control | 1.19 ± 0.08a | 0.77 ± 0.04a | 2.23 ± 0.24a | 23.05 ± 5.32a | 24.00 ± 2.96a |
| 12% PEG | 0.93 ± 0.10b | 0.46 ± 0.07b | 1.81 ± 0.21b | 21.62 ± 2.97b | 15.96 ± 1.85b |
| 18% PEG | 0.79 ± 0.07c | 0.44 ± 0.05c | 1.15 ± 0.40c | 17.33 ± 3.78c | 14.38 ± 2.93c |

*3.2. Metabolic Profile Analysis after Control and Drought Stress Treatments*

A total of 82 compounds, including sugars, sugar alcohols, amino acids, and organic acids, were annotated based on GC-MS results for the root, stem, and leaf tissue sample extracts analyzed after being subjected to different degrees of drought (Table S1). The PLS-DA model was created with two

principal components, which had explanatory and predictive values of 69.35%, 78.96%, and 63.68% for roots, stems, and leaves, respectively. The PLS-DA score plot employed three clusters that corresponded to different degrees of drought stress (Figure S1). Subsequently, many compounds, such as glycine, tyrosine, proline, aspartic acid, myo-inositol, sucrose, and fructose (vip > 1 and $p <$ 0.05), were identified as factors contributing to the distinct drought stress responses in different tissues. In this work, twenty compounds out of eighty-two, including eight sugars and sugar derivatives (myo-inositol, sucrose, glucose, fructose, mannose, galactose, and maltose) and twelve amino acids (valine, serine, threonine, alanine, aspartic acid, asparagine, glycine, proline, threonic acid, tyrosine, glutamic acid, and threose), were identified and are shown in Figures 3 and 4. Among sugars (Figure 3), the level of sucrose increased in both roots and leaves under water deficiency, whereas galactose was the highest in the 12%-PEG treatment group. A coordinated decrease in the accumulation levels of mannose was found in both the root and stem tissues under drought treatment (Figure 3b,e). On the other hand, one compound, glucose, revealed a decreasing trend in all tissues under drought stress. Maltose showed no significant changes in the roots among all experimental groups (p > 0.05). Changes in the amino acid patterns are shown in Figure 4. There were significant changes in the roots of drought-treated plants relative to the control, namely, changes in valine, serine, threonine, alanine, aspartic acid, and asparagine (Figure 4a). Specifically, a dramatic decrease was observed in the stem valine, serine, proline, threonic acid, and glycine contents under mild drought treatment, whereas the levels of these compounds increased significantly after severe drought stress (Figure 4f). Some components, including glycine, tyrosine, asparagine, aspartic acid, glutamic acid, threonic acid and serine, decreased considerably in leaves in response to drought stress (Figure 4f).

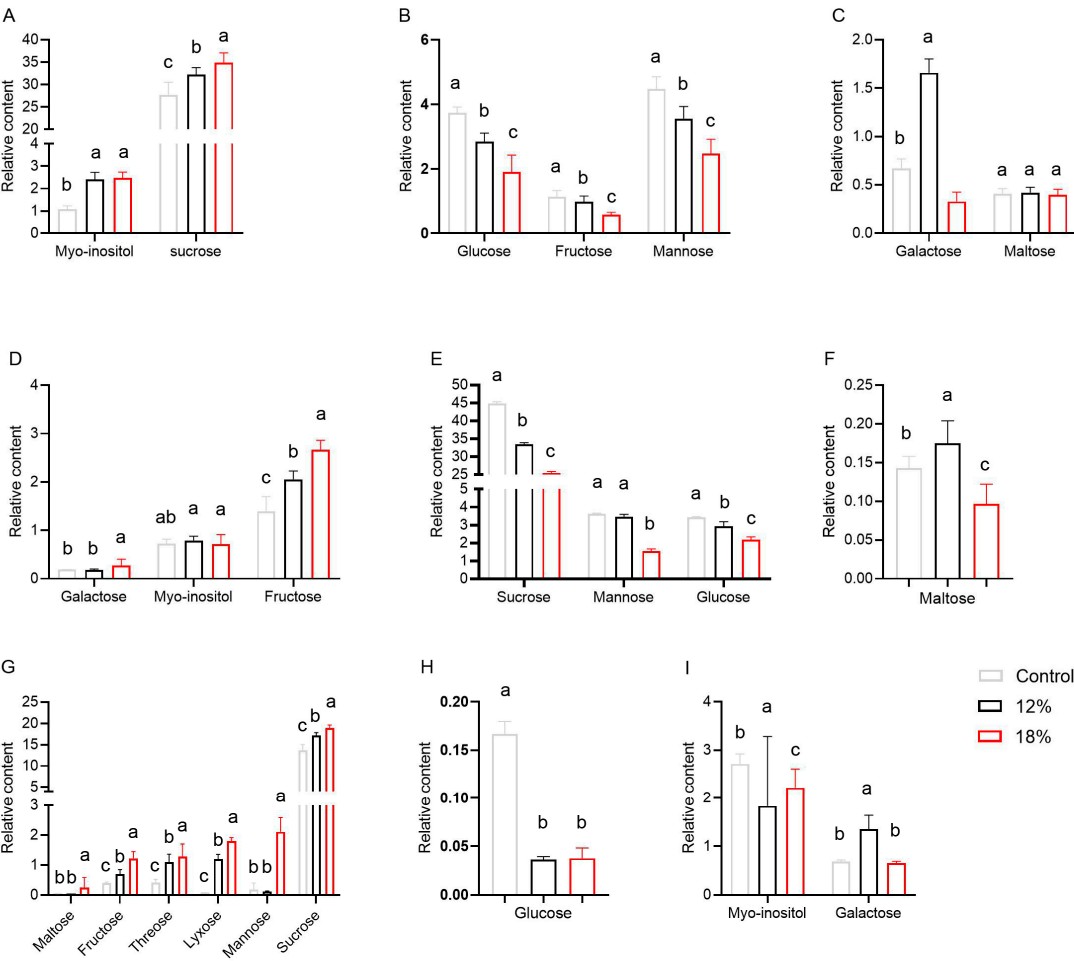

**Figure 3.** The relative contents of sugar metabolites in different tissues under the different treatments was analyzed using GC-MS (*n* = 3). (**A**), (**B**)**,** and (**C**) show the results in roots; (**D**), (**E**)**,** and (**F**) show the results in stems; and (**G**), (**H**), and (**I**) show the results in leaves. Each bar represents the mean ± SD (*n* = 3). Different lowercase letters above the columns indicate significant differences (*p* < 0.05).

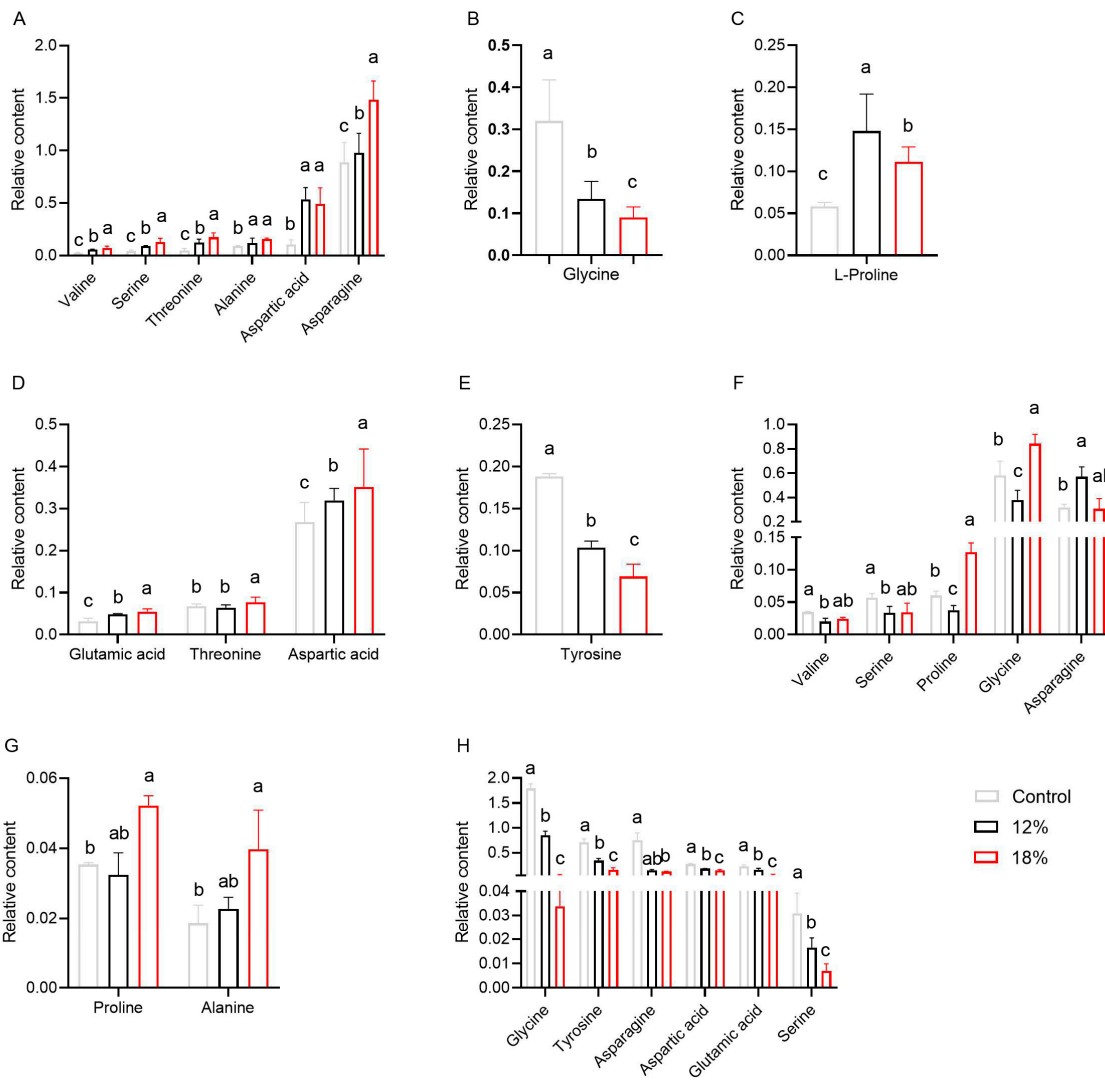

**Figure 4.** Amino acid contents in different tissues under different water conditions. (**A**), (**B**), and (**C**) show the results in roots; (**D**), (**E**)**,** and (**F**) show the results in stems; and (**G**) and (**H**) show the results in leaves. Each bar represents the mean ± SD (*n* = 3). Different lowercase letters above the columns indicate significant differences (*p* < 0.05).

### 3.3. Identification of Polyphenols in G. sinensis Adapted to Drought

This study represents the first comprehensive profiling of the 22 phenolic compounds that were tentatively identified from *G. sinensis* (Table S2), among which six were hydroxycinnamic acids or their derivatives (p-hydroxycinnamic acid, ferulic acid, caffeic acid, coumaric acid, cinnamic acid. and chlorogenic acid); thirteen were flavonols or their derivatives, such as isoliquiritigenin, glycyrrhizin, quercetin, abscisic acid, luteolin, kaempferol, rutin, naringin, genistein, apigenin, galangin, naringenin, and hesperetin; and two compounds with a carboxyl group attached to the aromatic ring were identified as simple phenolic metabolites, including syringic acid and benzoic acid. In addition, L-phenylalanine was identified, which is a phenolic compound that synthesizes major precursor compounds.

Further analysis was performed to determine the metabolic changes in the three tissues in response to drought stress. The resulting data are presented in a heatmap (Figure 5) with normalized sample values. The levels of genistein and galangin were significantly higher in the well-watered leaves and roots than in those in the treatment groups. Syringic acid, luteolin, and kaempferol accumulated primarily in the leaves and stems of the severe drought treatment group. These compounds showed different tissue-specific accumulation patterns under drought conditions. However, there were also some compounds, such as ferulic acid, with a similar trend in all tissues, with higher levels under moderate drought than under well-watered and severe drought conditions. In addition, the levels of some metabolites, including naringin, apigenin, and caffeic acid in leaves, abscisic acid and apigenin in roots, and chlorogenic acid and coumaric acid in stems, decreased with changes in the degree of drought.

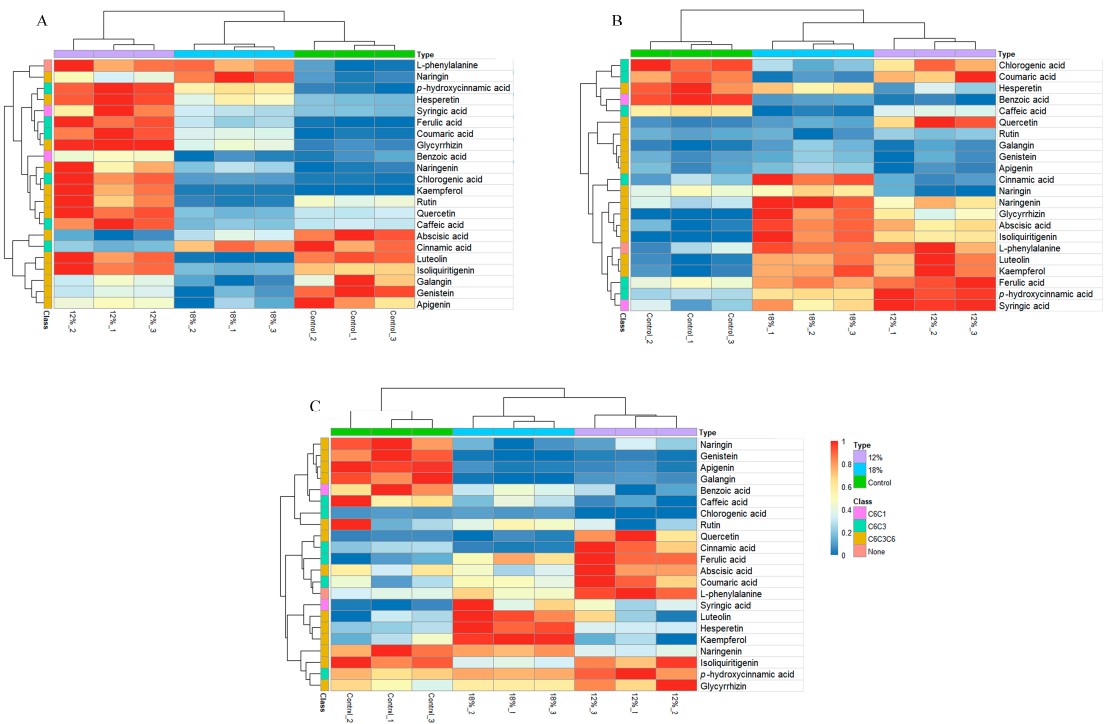

**Figure 5.** Hierarchical clustering of tissue-specific phenolic compounds in response to drought stress. The color key represents normalized values. Type: water status group; class: C6C1, C6C3C6, and C6C3 structures of phenolic metabolites. (**A**), (**B**), and (**C**) show the results in roots, stem and leaf

### 3.4. Phenotype, Physiological Parameters and Metabolites under Rehydration

To assess the changes in phenotype, physiological parameters, and metabolites in response to rehydration, three biological replicates of tissue samples were collected from plants following full rehydration and analyzed using nontargeted and targeted global metabolome technology based on GC-MS and UPLC-MS platforms, respectively. Drought caused the plants to grow slowly. In addition, the leaves turned yellow and wilted, the branches became scarce, and no new shoots were produced. During rehydration, in contrast, the yellow leaves were nearly completely restored to green, many new shoots were produced, and growth was resumed (Figure S2). In addition, some physiological parameters, such as the antioxidant enzymes CAT, APX, SOD, MDA, and POD, showed significantly lower levels than the original levels in the drought group (Figure S3). Additionally, the net photosynthetic rate, transpiration rate, stomatal conductance, and chlorophyll content increased (Figure S4). Many similarities were observed in the abundance of metabolites, including sugar, amino acid, and phenolic compounds. There were four, five, and eight amino acids

in the root, stem, and leaf tissues, respectively, and these compounds showed similar accumulation patterns in the rehydrated state and the well-watered state. Interestingly, the levels of asparagine and aspartic acid showed no change ($p > 0.05$) in all tissues between the control and rehydration groups (Figure 6a,c,e). However, various sugars that accumulated in response to rehydration remained at the same level in the control, with two-, five-, and four-sugar compounds distributed in the root, stem, and leaf tissues, respectively (Figure 6b,d,f). In comparing the primary compounds across all tissues in the rehydrated states, phenolic compounds, including ferulic acid, l-phenylalanine, galangin, genistein, apigenin, glycyrrhizin, and isoliquiritigenin, were detected, indicating that the relative contents were maintained ($p > 0.05$) in different tissues in the dry states, watered states, and fully rehydrated states (Figure S5).

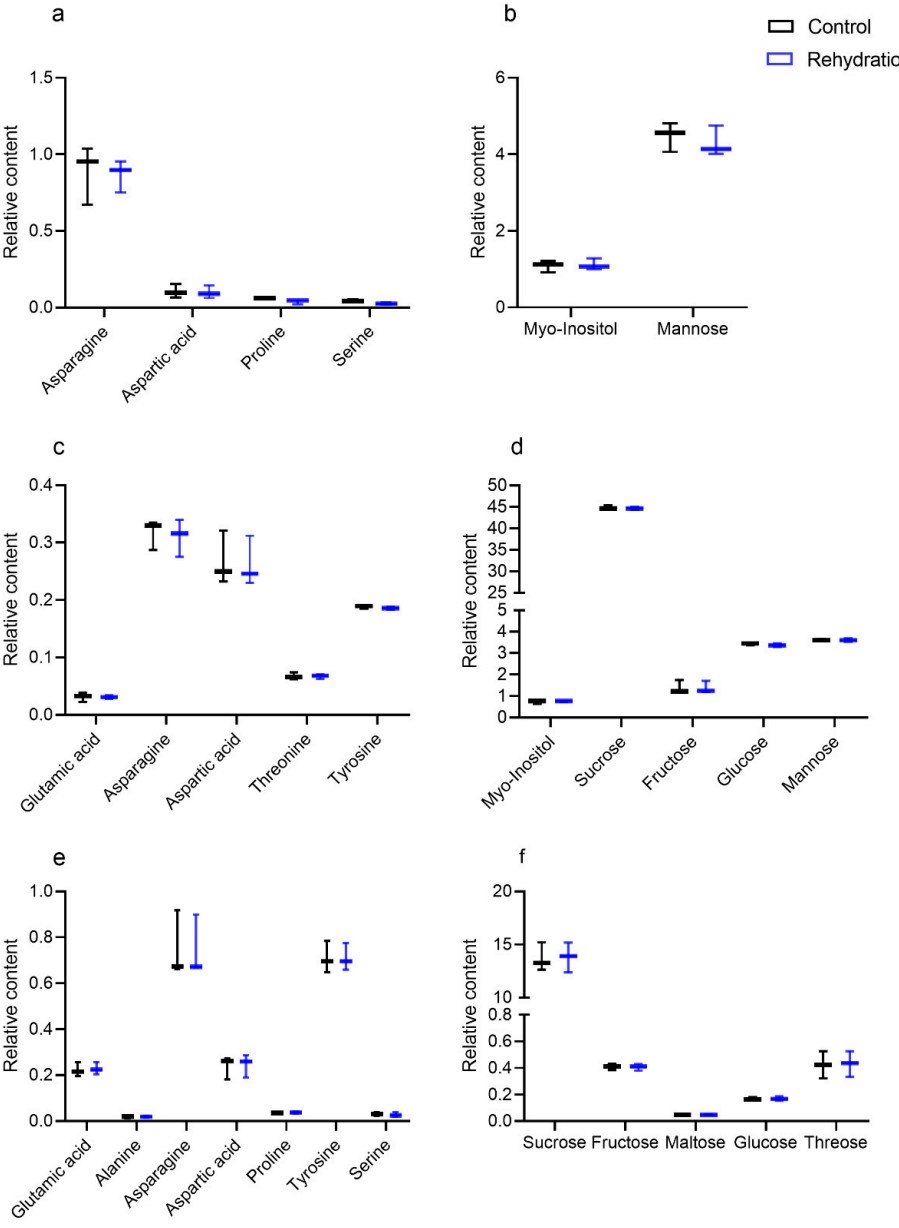

**Figure 6.** Comparison of compounds in the control and rehydration groups. The levels of amino acids are shown in (**a**), (**c**), and (**d**), which represent roots, stems, and leaves, respectively. Sugar compounds are shown in (**b**), (**d**), and (**f**), which represent roots, stems, and leaves, respectively. Significant differences were defined at $p < 0.05$.

## 4. Discussion

*Gleditsia sinensis*, also known as Chinese honey locust, soap bean, or soap pod, was historically used as a traditional Chinese medicine for its anti-inflammatory, antiseptic, and antitumor properties; moreover, this species has high drought tolerance [30]. With increasing environmental problems, drought stress is becoming an important factor influencing the growth of organisms. Therefore, comprehensive profiling of the chemical composition of *G. sinensis* in response to drought is necessary. SOD, POD, APX, and CAT are the four key enzymes of the antioxidant protection system, and changes in their activity directly show the protective reaction of plants in response to drought stress [31]. Our results showed that the SOD, POD, and CAT enzymes in active oxygen scavenging were significantly higher in the treatment groups than in the control. Moreover, the MDA content was significantly lower under drought conditions than under well-watered conditions, suggesting that drought-induced lipid peroxidation in the whole plant was more severe under severe drought stress. However, rehydration did not return these enzymes to their original state, indicating that the antioxidant enzyme system could not be completely repaired after the plant was subjected to severe drought stress. The chlorophyll content, stomatal conductance, and photosynthesis rate are important physiological indexes to measure the impact of environmental stress on plants [32]. Our research showed that the different index values in the leaves of *G. sinensis* were decreased with increasing drought degree, but this result was changed by rehydration (Figure S4), recovering the normal function of plant photosynthesis. Our results demonstrated that photosynthesis and water status are closely related, indicating that damage related to photosynthesis may be repaired when water conditions improve.

In response to drought stress, plants can alter their metabolites, such as amino acids and sugars, to cope with changes in water potential and to maintain turgor pressure [33]. According to our results, the major sugar observed in the aboveground tissues of plants was fructose, which accumulated markedly with increasing drought stress, whereas a significant decrease in fructose was observed in the roots. In contrast, the levels of glucose in different tissues showed a similar decreasing trend. In addition, the elevated levels of sucrose in both the roots and leaves of drought-treated plants were different from those in the stems (Figure 3). These soluble sugars are osmolytes, which function as osmoprotectants under water deficit. Our results show that metabolites exhibit different accumulation patterns in different tissues, regulating osmotic function and maintaining plant growth. Interestingly, a rehydration experiment was used to confirm the above conclusion, and some of the compounds (myo-inositol, fructose, glucose, sucrose, and mannose) returned to their original levels, indicating the function of these metabolites in phytoremediation. Moreover, some amino acid accumulation patterns in this study coincided with the observed sugar increases or decreases in different tissues after drought stress. In fact, osmotic adjustments and reactive oxygen species (ROS) detoxification are needed in water-stressed plants, and this demand is assisted by the resulting positive regulation of amino acids [34]. Proline, which is known to act as an osmoprotectant in plants suffering drought stress, accumulated in all tissues under severe drought treatment. Generally, the primary metabolite-backtracking mechanism was tissue-specific, and the levels of the compounds were significantly higher in aboveground tissues than in belowground tissues.

Phenolic accumulation is crucial to counteract the negative impacts of drought stress in plants [35]. Antioxidative phenolic metabolites can protect the enzymatic complexes present in different compartments of plant cells from the damaging effects of free radicals [36]. Comprehensive profiling of the phenolic compounds in *G. sinensis* was performed. Finally, twenty-two phenolic metabolites were identified, and their functional strategies were revealed in different tissues during drought stress (Table S2). Phenylalanine was identified as the amino acid involved in the synthesis of different classes of phenylpropanoid derivatives, which play an important role during defense against biotic and abiotic stresses [37]. Phenylalanine demonstrated a unified response strategy in all tissues, indicating that it has a conserved function in drought recovery. Hydroxycinnamic acids have a C6C3 structure with a double bond in the side chain in the cis-trans configuration. In this study, p-hydroxycinnamic acid, ferulic acid, caffeic acid, coumaric acid, cinnamic acid, and chlorogenic acid were identified and confirmed by comparison with standard compounds. These components showed a similar trend in the stem, with higher levels under drought stress than under well-watered

conditions. In addition, hydroxycinnamic acids have been reported to have antioxidant capacities and various biological activities [38]. However, the functions of these bioactive compounds in response to drought occurred mainly in the stem in our experiments. Thirteen compounds were identified with a C6C3C6 structure, including flavonols and their derivatives. Many flavonols, which can be found in various plants used in traditional medicine, have been reported to have potent antiangiogenic and antitumor activities [39]. Other studies have already demonstrated that these bioactive compounds accumulated in response to moderate stress. In our study, 12%-PEG treatment increased the levels of flavonols in roots. An important event in plants in response to drought stress is the triggering of abscisic acid (ABA) production. This phytohormone causes stomatal closure and regulates drought stress-related genes [40]; during our studies, the signal from this metabolite was recognized with LC-MS. This agrees with previous experiments in which the response to water stress and recovery included a significant increase in ABA synthesis in leaves. Moreover, abscisic acid contents peaked with stomatal closure and showed an opposite trend in leaves, which means that these factors have a negative relationship under water deficit conditions.

## 5. Conclusions

Water deficiency leads to a decrease in the enzymatic activity of antioxidants and a severe reduction in biomass. Moreover, antioxidant enzyme activity could not be completely restored in plant tissues following rehydration, while the capacity for photosynthesis could be recovered. Large amounts of amino acids and sugars were identified as involved in osmotic adjustment, partially compensating for the energy depletion under drought stress, while the resistance-related biosynthesis of amino acid and sugar metabolites satisfied the high demands resulting from vegetative growth and repair. In this study, a comprehensive identification of the phenolic compounds in this species was performed for the first time, in which 22 phenolic metabolites were identified, and their involvement in drought response was characterized.

**Supplementary Materials:** The following are available online at www.mdpi.com/xxx/s1, Figure S1: PLS-DA of primary metabolites of *G. sinensis*. whole samples, Figure S2: Phenotype of *G. sinensis* exposed to 18% PEG and rehydration, Figure S3: Comparison of SOD, POD, and CAT enzyme activities and MDA contents under control, 12% PEG, 18% PEG, and rehydration treatments, Figure S4: Photosynthetic parameters tin root, stem and leaf tissues of *G. sinensis* exposed to control, 12% PEG, 18% PEG and rehydration treatments, Figure S5: Hierarchical clustering of tissue-specific phenolic compounds in response to drought stress and rehydration. Table S1: Metabolites were identified by GC-MS, Table S2: Metabolite reporting checklist and recommendations for LC-MS.

**Author Contributions:** J.L. and ZH-T. conceived and designed the analysis and wrote and reviewed the paper. RK and LB-P collected the data and performed the experiment. Y.L. and KX-W. carried out analysis of the LC-MS and GC-MS, respectively. Y.S. and X.Y. wrote the original draft. J.L. and Y.L. reviewed the paper. All authors have read and agreed to the published version of the manuscript.

**Funding:** This study was financially supported by the Central Universities in China (2572017DA05), the China Postdoctoral Science Foundation (415638), the Heilongjiang Provincial Postdoctoral Science Foundation (415560), and the Heilongjiang Provincial Postdoctoral Science Foundation (415556).

**Conflicts of Interest:** The authors declared that they have no conflicts of interest to this work. We declare that we do not have any commercial or associative interest that represents a conflict of interest in connection with the work submitted

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
