# Peer review of "Differential Metabolite Accumulation in Different Tissues of Gleditsia sinensis under Water Stress and Rehydration Conditions"

_forests, doi:10.3390/f11050542_

Round 1
Reviewer 1 Report
This is a well performed study, that deserves publication after some considerations.
The major flaw I found in the design to induce water stress in plants. I have read several papers trying to induce chronic stress by using PEG, so no problem with that. However, I really consider that the study would improve a lot if authors could provide data of the "real" water stress suffered by each treatments. Do you have any estimation of water status in treatment 2 and 3? Water content or water potential would be very wellcome in this sense. Of course, symptoms of water stress are evidents: plant morphology and gas exchange indicate that...but with the present structure of data we cannot really know the extend and severity of that.
This is particularly important if we consider the induction of xylem cavitation by water stress. In such a case, the recovery of the whole plant performance is not absolutely achieved. In this sense, I recommend to consider the following paper:
- Peguero-Pina JJ et al (2018) Cavitation Limits the Recovery of Gas Exchange after Severe Drought Stress in Holm Oak (Quercus ilex L.)Forests 2018, 9(8), 443; https://doi.org/10.3390/f9080443
Here the authors showed that metabolic (s.l.) functionality can be easily and rapidly recovered after severe drought. However, once reached a drastic reduction in hydraulic conductivity by cavitation, the overall performance is affected.
Authors should take this into account, and discuss the results taking this limitation in mind.
I find the chemical results as very interesing and sound, and this is a major strength of this paper. However, authors show relative importance. Should it possible to show values in concentration by dry mass of the plant?
I have not detect major problems with the english, but a few mistakes. As an example,
"Metabolites showed a significant increase or decrease under drought stress were considered to be the key metabolites of interest for a better understanding of the drought tolerance mechanisms." Probably you wanted to say:...Metabolites showing..."
There are a few more typographyc mistakes that will be detected by authors with a deep reading of their own manuscript (eg. sponins instead of saponins)
Author Response
Dear Reviewer,
We are grateful to your helpful comments and thoughtful suggestions on our manuscript entitled “Role of chemical components of different tissues of Gleditsia sinensis under water stress and rehydration conditions”. We have extensively revised the manuscript according to the suggested comments point by point. The modified parts were highlighted in the text with a red word. The detailed responses to the referees are as follows:
Comment 1: The major flaw I found in the design to induce water stress in plants. I have read several papers trying to induce chronic stress by using PEG, so no problem with that. However, I really consider that the study would improve a lot if authors could provide data of the "real" water stress suffered by each treatment. Do you have any estimation of water status in treatment 2 and 3? Water content or water potential would be very well come in this sense. Of course, symptoms of water stress are evidents: plant morphology and gas exchange indicate that...but with the present structure of data we cannot really know the extend and severity of that.
Responses: Thanks for your comments. PEG-6000 solutions of different water potentials are prepared according to the formula, which is as follows:
ψs=-(1.18×10-2) C-(1.18×10-4) C2+(2.67×10-4)CT+(8.39×10-7)C2T -----------[1]
Among them, ψs --the water potential of the solution; C- the concentration of PEG-6000; T-- temperature, ℃.
Therefore, the water potential of the 12% PEG-6000 treatment group was -0.2MPa, and the water potential of the 18% PEG-6000 treatment group was -0.4MPa. In order to show the "real" water stress suffered by each treatment, the water potential data has been added in Materials and Methods.
[1] B.E. Michel, M.R. Kaufmann, The Osmotic Potential of Polyethylene Glycol 6000, Plant Physiology 51(1973) 914-916.
Comment 2: I find the chemical results as very interesing and sound, and this is a major strength of this paper. However, authors show relative importance. Should it possible to show values in concentration by dry mass of the plant?
Responses: Thanks for your comments. Displaying the concentration value by dry mass of the plant can show the actual content of the compound indeed, but this article shows the relative level mainly because of the following two points: Firstly, we focus on changes in chemical composition more, not on the actual content of compounds. Secondly, because of the principle of compound identification in metabolomics, the use of relative content is more suitable for data processing and is also widely used [2, 3].
[2] L. Wang, T. Nagele, H. Doerfler, L. Fragner, P. Chaturvedi, E. Nukarinen, A. Bellaire, W. Huber, J. Weiszmann, D. Engelmeier, Z. Ramsak, K. Gruden, W. Weckwerth, System level analysis of cacao seed ripening reveals a sequential interplay of primary and secondary metabolism leading to polyphenol accumulation and preparation of stress resistance, Plant J(2016).
[3] A. Moing, A. Aharoni, B. Biais, I. Rogachev, S. Meir, L. Brodsky, J.W. Allwood, A. Erban, W.B. Dunn, L. Kay, S. de Koning, R.C. de Vos, H. Jonker, R. Mumm, C. Deborde, M. Maucourt, S. Bernillon, Y. Gibon, T.H. Hansen, S. Husted, R. Goodacre, J. Kopka, J.K. Schjoerring, D. Rolin, R.D. Hall, Extensive metabolic cross-talk in melon fruit revealed by spatial and developmental combinatorial metabolomics, New Phytol 190(2011) 683-696.
Comment 3: I have not detect major problems with the english, but a few mistakes. As an example,
"Metabolites showed a significant increase or decrease under drought stress were considered to be the key metabolites of interest for a better understanding of the drought tolerance mechanisms." Probably you wanted to say:...Metabolites showing..."
There are a few more typographyc mistakes that will be detected by authors with a deep reading of their own manuscript (eg. sponins instead of saponins)
Responses: Thank you for pointing this out. In the revised version, we have modified these English mistakes.
Reviewer 2 Report
The manuscript "Role of chemical components of different tissues of Gleditsia sinensis under water stress and rehydration conditions" describes the changes in metabolite composition in chinese soap pod plant under water stress conditions. The manuscript is well written. I recommend the manuscript for publication with a couple minor revisions:
- The title does not reflect the nature of this work. The manuscript describes differential metabolite accumulation in the above said plant under stress conditions, but do not provide direct proof of causal relationship between metabolite composition and stress physiology. Nevertheless, the manuscript endows important observation in terms of , and I suggest the authors modify the title to : "Differential metabolite accumulation in different tissues of...".
- Check the english to correct the use of numbers to start a sentence- such as "8 sugars, 12 amino acids and 22 phenolic compounds, were characterized- should read as- Eight sugars…"
Author Response
Dear Reviewer,
We are grateful to your helpful comments and thoughtful suggestions on our manuscript entitled “Role of chemical components of different tissues of Gleditsia sinensis under water stress and rehydration conditions”. We have extensively revised the manuscript according to the suggested comments point by point. The modified parts were highlighted in the text with a red word. The detailed responses to the referees are as follows:
Comment 1: The title does not reflect the nature of this work. The manuscript describes differential metabolite accumulation in the above said plant under stress conditions, but do not provide direct proof of causal relationship between metabolite composition and stress physiology. Nevertheless, the manuscript endows important observation in terms of, and I suggest the authors modify the title to: "Differential metabolite accumulation in different tissues of...".
Responses: Thanks for your kind reminder and comment. The manuscript title has been modified.
Comment 2: Check the English to correct the use of numbers to start a sentence- such as "8 sugars, 12 amino acids and 22 phenolic compounds, were characterized- should read as- Eight sugars…"
Responses: Thank you for pointing this out. We have modified the use of numbers to start a sentence.
Round 2
Reviewer 1 Report
After the modifications done by the authors and responses, I suggest to accept this manuscript. As a plant physiologist, I would like to know concetration of the different compounds (in any basis, as dry mass...). However, authors reply that it is commonly used this presentation, in form of relative increments. So, I do not insist on this, as the results have to be used by scientist with a similar focus.
The very minor language mistakes have been incorporated, so I do not have more suggestions in this sense also.
Author Response
Dear Reviewer,
Thank you for your comments and suggestions for this manuscript、
Kind regards, Jia Liu